# Aligning with Human Judgement: The Role of Pairwise Preference in Large Language Model Evaluators

**Yinhong Liu**[1*]  **Han Zhou**[1*]  **Zhijiang Guo**[1]  **Ehsan Shareghi**[2]  **Ivan Vulić**[1]
**Anna Korhonen**[1]  **Nigel Collier**[1]
[1]University of Cambridge  [2]Monash University
{yl535,hz416,zg283,iv250,alk23,nhc30}@cam.ac.uk  ehsan.shareghi@monash.edu

## Abstract

Large Language Models (LLMs) have demonstrated promising capabilities as automatic evaluators in assessing the quality of generated natural language. However, LLMs still exhibit biases in evaluation and often struggle to generate coherent evaluations that *align* with human assessments. In this work, we first conduct a systematic study of the misalignment between LLM evaluators and human evaluation, revealing that existing calibration methods aimed at mitigating biases of LLMs are insufficient for effectively aligning LLM evaluators. Inspired by the use of preference data in RLHF, we formulate the evaluation as a ranking problem and introduce **Pair**wise-preference **S**earch (PAIRS), an uncertainty-guided search method that employs LLMs to conduct pairwise comparisons *locally* and efficiently ranks candidate texts *globally*. PAIRS achieves state-of-the-art performance on representative evaluation tasks in long-form generations and demonstrates significant improvements over direct scoring. Furthermore, we provide insights into the role of pairwise preference in quantifying the transitivity of LLMs and demonstrate how PAIRS benefits from calibration using debiased pairwise evaluations.

## 1 Introduction

Large Language Models (LLMs) (Brown et al., 2020; OpenAI, 2023; Anil et al., 2023a;b) have achieved promising instruction-following abilities, enabling an efficient adaptation pipeline to solve complex natural language tasks with minimal or no labelled data (Kojima et al., 2022; Zhou et al., 2024c). This unique capability stems from supervised training LLMs with instruction-following data (Wei et al., 2022; Chung et al., 2022) and reinforcement learning from human feedback (RLHF) (Christiano et al., 2017; Stiennon et al., 2020). In the RLHF training paradigm, a reward model is aligned with human preferences based on ranked comparison data (Ouyang et al., 2022). It enhances LLMs' alignment with human values (Touvron et al., 2023), thereby generating responses that better assist humans and abide by human values (Dai et al., 2024).

Recently, LLMs have emerged as competent and inexpensive assessors for the evaluation of various aspects, such as coherence, fluency, consistency and hallucination, of text quality (Chen et al., 2023; Zeng et al., 2024; Zheng et al., 2024). Put simply, LLMs can be seen as an appealing reference-free evaluation paradigm for generative tasks while avoiding expensive human-labeling costs (Liu et al., 2023a). However, the predictions from LLM evaluators are highly sensitive to the prompt design (Zhou et al., 2023; Zheng et al., 2024) and even biased by multiple sources, including positional bias (Wang et al., 2023b), verbosity bias (Saito et al., 2023), and contextual bias (Zhou et al., 2024b). These biases still hinder fair and trustworthy applications of LLM evaluators, resulting in inconsistency and misalignment with human judgements.

To mitigate biased predictions from LLMs, calibration techniques have been developed to ensure less biased outcomes (Liu et al., 2023b; Li et al., 2023a; Zhou et al., 2024b). Motivated

---

*Equal contribution. Code is available at https://github.com/cambridgeltl/PairS.

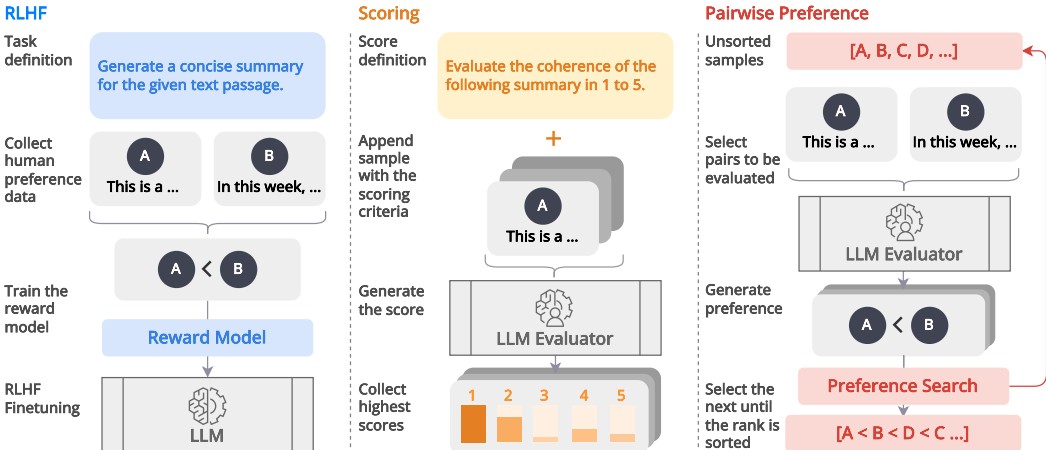

Figure 1: Illustration of RLHF (left), direct scoring with LLM evaluators (middle), and pairwise preference search (PAIRS; right). Pairwise preference data have been utilized to train the reward model to align the LLM in RLHF. Leveraging this idea, PAIRS reframes the traditional scoring-based evaluations into a series of pairwise comparisons between selected candidate pairs which are then used to form the final ranking.

by this line of work, we first conduct a systematic analysis of the effectiveness of calibration techniques in aligning LLM evaluators in direct scoring (Figure 1, middle) with human judgements. We find that existing calibration methods are insufficient for aligning LLM evaluators, even when provided with supervised data. We discover that misalignment in evaluation is not primarily due to biased priors over evaluation scores, but rather from misaligned evaluation standards, i.e., the likelihood of LLM evaluators defined in Equation (1). Therefore, we are motivated to explore a new evaluation paradigm with LLMs to facilitate a more aligned judgement.

Inspired by how LLMs are aligned through a reward model with preference data in RLHF, we argue that LLM evaluators can produce more aligned and inherent evaluations by leveraging their pairwise ranking abilities. Recent work has started exploring this intrinsic property by prompting LLMs with a pair of candidates (Wang et al., 2023b; Chen et al., 2023; Qin et al., 2023; Li et al., 2024). However, the complexity and scalability of evaluating with pairwise preference have been largely overlooked. To produce a preference ranking for evaluation, existing methods require exhaustive pairwise comparisons, as they ignore the transitivity assumption[1] (Qin et al., 2023). This makes the evaluation procedure intractable as the number of comparisons grows quadratically (Zheng et al., 2024; Liusie et al., 2024).

In this work, we shed new light on the full potential of pairwise preference in LLM evaluators and its alignment with human judgements. We propose **Pair**wise-preference **S**earch (PAIRS), an uncertainty-guided search method that efficiently estimates the Maximum Likelihood Estimate (MLE) of preference rankings by searching through an uncertainty-pruned pairwise comparison space. PAIRS is more effective and query-efficient than existing pairwise baselines while substantially outperforming traditional direct scoring evaluations. We extensively examine the applicability of PAIRS in representative evaluation tasks, including summarization and open-ended generation. We show that PAIRS produces more robust and transitive evaluations compared with the greedy search baseline. Lastly, we provide further insights into how pairwise preference can be used to quantify the transitivity ability of LLM evaluators and how it benefits from calibration.

In sum, we provide the following contributions: **1)** We present a systematic analysis of the limitations of calibration in aligning direct scoring LLM evaluators with human judgements. **2)** We formulate the evaluation as ranking from the perspective of transitivity and propose

---

[1]The term transitivity in this work refers to mathematical transitivity, not linguistic transitivity. It is a property stating that if A is preferred to B and B is preferred to C, then A is preferred to C.

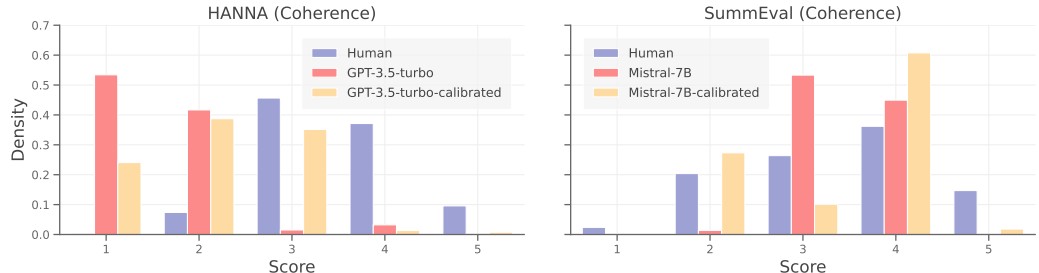

Figure 2: *LLM evaluations are misaligned with human judgements.* The score histograms on evaluating the coherence in HANNA (Chhun et al., 2022) and SummEval (Fabbri et al., 2021). We present the scores from gold human evaluations, LLMs, and LLMs after calibrations. The histograms can be interpreted as estimated score prior distributions via marginalization.

PAIRS, an uncertainty-guided pairwise preference search method that efficiently estimates MLE rankings. **3)** We demonstrate that PAIRS attains unique scalability in aligning LLM evaluations. We also provide insights into the transitivity and calibration of pairwise preferences in LLM evaluators.

## 2  On the Limitations of Calibration in Aligning LLM Evaluators

**Misalignment Between LLM Evaluators and Human Judgements.** The prediction of score-based LLM evaluators can be expressed as a posterior predictive distribution $p_{\mathcal{M}}(s|y; x, I)$:

$$p_{\mathcal{M}}(s|y; x, I) \propto \underbrace{p_{\mathcal{M}}(y|s; x, I)}_{Likelihood} \times \underbrace{p_{\mathcal{M}}(s; x, I)}_{Prior}, \tag{1}$$

where $x$ and $y$ form an input-output pair (e.g., a source text and its summary, respectively), $I$ denotes an instruction prompt, and $s$ represents the model's rating score in its evaluation. Similarly, we can express the human posterior predictive distribution as $p_{\mathcal{H}}(s|y; x, I)$. For simplicity, we omit the $x$ and $I$ terms hereafter. When utilizing LLMs to perform Likert scale evaluations (Likert, 1932), it is known that there exists a mismatch between the score distributions of LLM ratings and human annotations (Liu et al., 2023b). We start by observing and interpreting the same misalignment phenomenon, as shown in Figure 2.

Since the posterior distribution consists of two components, the likelihood, and the score preference prior, we hypothesise that *the discrepancy between the posterior distributions of LLMs and humans mainly arises from a misalignment between their respective likelihoods.* To verify this hypothesis, we conduct a preliminary experiment by calibrating the LLM's score preference prior, $p_{\mathcal{M}}(s)$ to match the human prior, $p_{\mathcal{H}}(s)$, described in what follows.

**Score-Based LLM Evaluator Calibration.** Following previous calibration methods (Zhou et al., 2024b; Jiang et al., 2023b), we first assume that the likelihood terms for both the LLM and humans are the same, $p_{\mathcal{M}}(y|s) = p_{\mathcal{H}}(y|s)$. Upon this assumption, the human posterior distribution $p_{\mathcal{H}}(s|y)$ can be expressed as:

$$p_{\mathcal{H}}(s|y) = \frac{p_{\mathcal{H}}(s)p_{\mathcal{H}}(y|s)}{p_{\mathcal{H}}(y)} \propto \frac{p_{\mathcal{H}}(s)}{p_{\mathcal{M}}(s)} p_{\mathcal{M}}(s|y). \tag{2}$$

We can estimate LLM's preference prior $p_{\mathcal{M}}(s)$ by marginalizing the model's predictions, i.e., $p_{\mathcal{M}}(s) = \sum_y p_{\mathcal{M}}(s|y)p(y)$. Unlike previous work that assumes a uniform prior for $p_{\mathcal{H}}(s)$ (Zhou et al., 2024b), we directly estimate it by marginalizing human annotations to achieve a more accurate estimation. The calibration is performed by scaling the model's predictions according to Equation (2).

In Figure 2, the experimental results show that *even after calibration, the gap persists between the posterior distributions of the LLM and human evaluators.* Specifically, the Mean Absolute

Error (MAE) shifts from 1.62 to 1.16 on HANNA and from 0.78 to 0.86 on SummEval, indicating notable misalignment. This suggests that the discrepancy is *not* primarily due to the misaligned priors but rather stems from the differences in the likelihood of LLM evaluators. In other words, the assumption that LLM evaluators and humans have similar evaluation likelihoods becomes invalid, i.e., $p_{\mathcal{M}}(y|s) \neq p_{\mathcal{H}}(y|s)$. The likelihood term reflects expected output candidates for a given score, which can be interpreted as the underlying evaluation standards. LLMs learn their standards from pretraining data, potentially diverging from the human evaluation standards. Hence, these findings motivate efforts to reduce the gap between evaluation standards.

## 3 Uncertainty-Guided Pairwise-Preference Search

### 3.1 Evaluation as Ranking

Inspired by the fact that the RLHF training stage of most recent LLMs utilizes pairwise preference or preference ranking data (Touvron et al., 2023), we propose that evaluation with pairwise comparisons could be a promising solution to address the above issues. We conclude the motivation: *The difference between LLM and human evaluation standards is smaller when performing pairwise comparison, compared to rating with scores.*

When using an LLM to perform a pairwise comparison, we can compute the preference probabilities $P(y_i \succ y_j)$ from output logits, where $\succ$ denotes the preference relation "is preferred to". However, LLM evaluators do not guarantee transitivity property, e.g. given an LLM's pairwise preferences $y_i \succ y_j$ and $y_j \succ y_k$, it is not necessarily true that the LLM will have $y_i \succ y_k$. The problem of finding the optimal preference ranking from non-transitive pairwise comparison probabilities is known as the Rank Aggregation problem (Dwork et al., 2002). We formulate this problem as a Maximum Likelihood Estimation (MLE) problem.

Let $y_{1:N}$ be a set of $N$ output candidates to be ranked. For a pair of outputs $(y_i, y_j)$, we have the probability $P(y_i \succ y_j)$ that output $y_i$ is preferred over output $y_j$. We denote a permutation (ranking) of these candidates as $\pi = (\pi_1, \pi_2, \ldots, \pi_N)$, where $\pi_i$ is the index of the $i$-th ranked candidate, i.e., $y_{\pi_i}$ is the $i$-th ranked candidate output in $\pi$. The ranking $\pi$ imposes a total order on $y_{1:N}$, such that $y_{\pi_1} \succ y_{\pi_2} \succ \ldots \succ y_{\pi_N}$. The goal is to find a ranking of the output candidates that maximizes the likelihood of observing pairwise probabilities:

$$\mathcal{L}_{NT}(\pi) = \prod_{i,j:\pi_i < \pi_j} P(y_{\pi_i} \succ y_{\pi_j}), \tag{3}$$

where $\mathcal{L}_{NT}$ stands for the non-transitive likelihood,[2] which does not make any transitive assumptions for the pairwise comparison probabilities. The MLE ranking of $\mathcal{L}_{NT}$ is also equivalent to the Kemeny-Optimal Ranking (Kemeny, 1959). However, finding the optimal ranking is challenging in two aspects:

1. Computing $\mathcal{L}_{NT}$ requires a complete set of $O(N^2)$ pairwise comparisons.
2. The exact solution to the rank aggregation problem is NP-hard (Dwork et al., 2001), meaning that there is no known polynomial-time algorithm to solve it optimally.

If LLM evaluators are assumed to possess transitivity property, then the problem of finding the MLE ranking reduces to a sorting problem, which can be solved by standard sorting algorithms such as merge sort or quick sort with only $O(N \log N)$ pairwise comparisons.

However, strict transitivity is a strong assumption. To relax the strict transitivity assumption to some extent and find a better trade-off between computational complexity and performance, we propose simplifying the non-transitive likelihood in Equation (3) by assuming compositional stochastic transitivity (Latta, 1979; Oliveira et al., 2018). The derivation of this simplified likelihood is provided in Appendix §C.2:

---

[2]This likelihood can be derived from the general learning to rank likelihood in Appendix §C.1.

**Algorithm 1** Uncertainty-Guided PAIRS-beam

1: **Objective**: Find the ranking that maximizes the transitive likelihood $\mathcal{L}_T$.
2: **Hyper-parameters**: Beam size, $beam\_size$; Uncertainty threshold, $U_h$.
3: **Inputs**: Two sorted input sub-arrays $\mathbf{a} = [A_1, A_2, \ldots, A_L]$ and $\mathbf{b} = [B_1, B_2, \ldots, B_R]$.
4: **Output**: A merged sorted array.
5: **Initialize**: Initialize a Beam $\mathcal{B}$ with the first candidate *cand*. Each *cand* keeps track of its choice trajectory, two pointers $i$ and $j$ for $\mathbf{a}$ and $\mathbf{b}$, and a cumulative likelihood of its trajectory.
6: **for** $k = 1$ to $L + R$ **do**
7:     Create another empty beam $\mathcal{B}'$.
8:     **for** *cand* in $\mathcal{B}$ **do**
9:         $P(A_i \succ B_j), U_{A_i, B_j} \leftarrow \text{LLM}(A_i, B_j)$
10:         **if** $U_{A_i, B_j} > U_h$ **then**
11:             $\mathcal{B}' \leftarrow \mathcal{B}' \cup cand.\text{update}(A_i)$
12:             $\mathcal{B}' \leftarrow \mathcal{B}' \cup cand.\text{update}(B_j)$
13:         **else if** $P(A_i \succ B_j) \geq \frac{1}{2}$ **then**
14:             $\mathcal{B}' \leftarrow \mathcal{B}' \cup cand.\text{update}(A_i)$
15:         **else**
16:             $\mathcal{B}' \leftarrow \mathcal{B}' \cup cand.\text{update}(B_j)$
17:         **end if**
18:     **end for**
19:     $\mathcal{B}' \leftarrow \text{sort}(\mathcal{B}', \text{key} = \mathcal{L}_T)$
20:     $\mathcal{B} \leftarrow \mathcal{B}'[: beam\_size]$
21: **end for**
22: **return** $\mathcal{B}[0].\text{trajectory}$

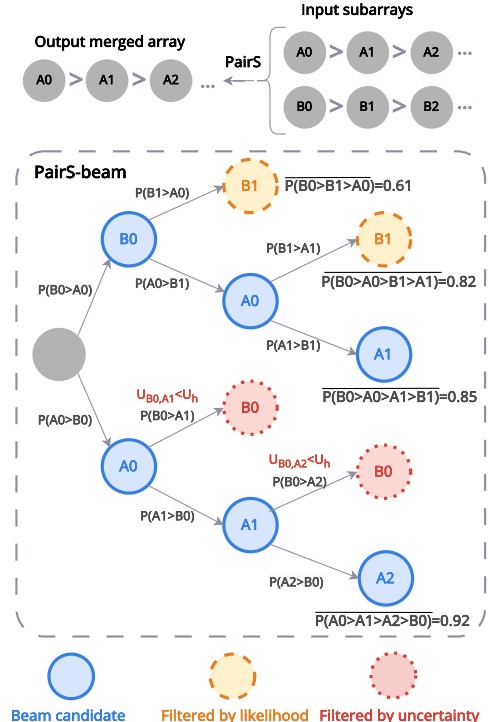

Figure 3: Visualization of PAIRS in ranking with a beam size of 2 on the same evaluation examples in Figure 4.

$$\mathcal{L}_T(\pi) = \prod_{\pi_i < \pi_{i+1}} P(y_{\pi_i} \succ y_{\pi_{i+1}}), \tag{4}$$

where $\mathcal{L}_T$ stands for the transitive likelihood, and the MLE ranking is denoted as $\pi^* = \arg\max_\pi \mathcal{L}_T(\pi)$.

## 3.2 Uncertainty-Guided Pairwise-Preference Search

The traditional merge sort algorithm employs a divide-and-conquer approach, merging two sorted subarrays at each operation step. This merge operation can be interpreted as a greedy search procedure that aims to find the MLE ranking $\pi^*$ under the transitive likelihood. We refer to this approach as **PAIRS-greedy**. However, the non-transitive nature of LLMs suggests that optimization based on local probability may not lead to the optimal solution. Therefore, we also propose **PAIRS-beam**, a heuristic beam search algorithm that improves upon the greedy search merge operation by considering both candidates at each step.

The PAIRS algorithm is described in Algorithm 1, and a demonstration of its workings is provided in Figure 3 and Figure 4. Intuitively, while the PAIRS-greedy chooses the preferred items between the heads of two sub-arrays in a greedy fashion, PAIRS-beam main-

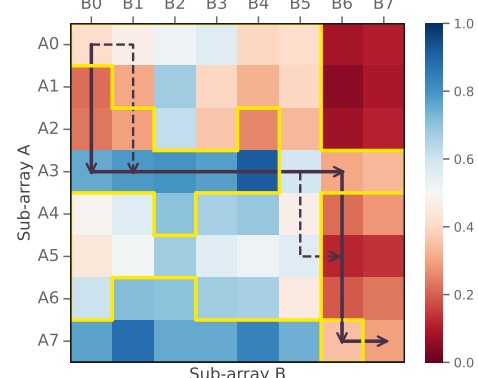

Figure 4: Illustration of PAIRS with a beam size of 2. Heatmap values show $P(A_i \succ B_j)$ by Mistral 7B, deciding the search direction. Yellow-bordered lines highlight comparisons whose uncertainties are above $U_h$. **Solid lines** represent greedy search paths, while dashed lines denote alternative paths by other beam candidates.

tains a likelihood $\mathcal{L}_T$ beam that keeps track of the top-k trajectory candidates. By also considering less preferred items between the two heads, PAIRS explores a broader search space and potentially finds a better ranking.

To further improve the efficiency of the beam search, we introduce a pruning mechanism based on the LLM's uncertainty. When the uncertainty of the LLM's pairwise comparison is lower than a threshold $U_h$, we prune the less preferred candidate from the beam. This uncertainty-based pruning restricts the search space, making the beam search more efficient and achieving a better balance between computational complexity and ranking performance.

To represent the uncertainty between two items, we follow a commonly used approach in previous works (Wan et al., 2023a;b) that employs entropy as the uncertainty measurement. Hence, the uncertainty between a pair of comparisons can be expressed below:

$$U(y_i, y_j) = H(y_i, y_j) = -P(y_i \succ y_j) \log(P(y_i \succ y_j)) - P(y_j \succ y_i) \log(P(y_j \succ y_i)). \quad (5)$$

### 3.3 A Scaling Variant of PAIRS

The theoretical complexity of PAIRS-beam ranges from $O(N \log N)$ to $O(N^2)$ pairwise comparisons, as shown in Figure 5. The upper bound occurs when the uncertainty-based pruning is disabled, and the beam size is sufficiently large to explore all potential trajectories during the merge operation. Conversely, the lower bound corresponds to PAIRS-greedy. However, for large instances, even the complexity of $O(N \log N)$ becomes computationally expensive and infeasible. To address this issue, we propose a two-stage scaling method that leverages the granularity of Likert scale annotations.

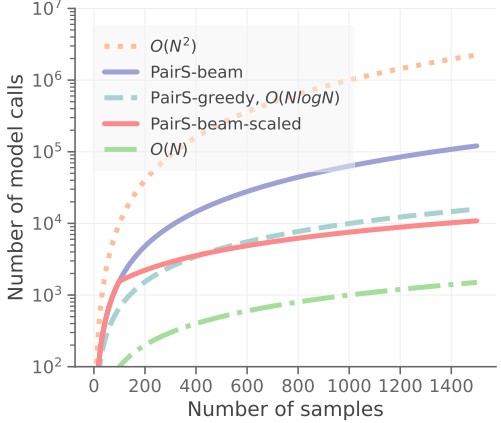

Figure 5: PAIRS *is scalable.* Complexity plot for different pairwise preference algorithms.

In the first stage, we sample and rank a subset of the evaluation set, referred to as the anchor set. The size of the anchor set can be statistically determined by accommodating a tolerable error margin (Cochran, 1977) (Further details and empirical evidence in Appendix §F.3). In the second stage, we use binary search to find the positions of the remaining samples relative to the anchor set, efficiently determining their appropriate positions in the overall ranking. As depicted in Figure 5, the second stage of the scaling method results in $O(N)$ complexity, a significant improvement over the original complexity. This two-stage scaling approach allows PAIRS to efficiently handle large evaluation sets while maintaining the benefits of beam search and uncertainty-based pruning. The scaling variant strikes a balance between complexity and ranking performance, making it suitable for real-world applications involving large-scale datasets.

### 3.4 On the Benefits of Pairwise-Preference Search

**Transitivity.** Both PAIRS-direct and PAIRS-beam assume transitivity to some extent, implying that an LLM with better transitivity can perform more accurate and more efficient evaluations. In other words, we expect better-performing LLMs to exhibit more transitive behaviour when making pairwise comparisons. Consequently, we propose that transitivity can serve as an effective measure of an LLM's ability as an evaluator. By quantifying and comparing the transitivity of different LLMs, we can gain insight into their relative performance and suitability for evaluation tasks. To validate this hypothesis, we provide complementary experiments and analysis in Section 5.4.

**Calibration.** Although pairwise preference inherently aligns LLM evaluations with human judgement, pairwise prompting can still suffer from misaligned preference priors, e.g.,

positional bias and contextual bias (Zhao et al., 2021; Zhou et al., 2024b). Therefore, we propose applying calibration on top of PAIRS to ensure debiased PAIRS evaluations. It is worth noting that, unlike calibrating the score-based LLM evaluators, which requires external information on the distribution of human preference prior, as shown in Section 2, pairwise comparisons can fully leverage a uniform human prior, due to the random nature of pairwise sample selections.

## 4    Related Work

**LLMs as Evaluators.** Recently, LLMs have been increasingly used to automatically evaluate various natural language generation tasks (Zhong et al., 2022; Chen et al., 2023; Wang et al., 2023a; Tan et al., 2024). LLM evaluators enable reference-free or reference-guided metrics while being cheaper alternates to human evaluation in a diverse task set, such as summarization (Shen et al., 2023), machine translation (Kocmi & Federmann, 2023), open-ended story generation (Chiang & Lee, 2023), dialog response generation (Liu et al., 2024a) and instruction-following (Zeng et al., 2024). Unlike traditional reference-based evaluation metrics (Papineni et al., 2002; Liu et al., 2024b), LLM evaluators are usually equipped with evaluation criteria prompts and generate scores to access the quality of the candidate text (Liu et al., 2023a). This forms the scoring-based LLM evaluator that enables easy adaption to various tasks by simple instructions (Fu et al., 2023). To fairly examine various LLM evaluation abilities, efforts have also been devoted to developing meta-evaluation benchmarks (Wang et al., 2023b; Zhang et al., 2023; Zheng et al., 2024) and prompt optimization (Zhou et al., 2024a).

**Pairwise Preference.** Preference data have been widely used in training the reward model to reflect human preferences in RLHF (Ouyang et al., 2022). For example, Llama-2 has leveraged binary preference with 'chosen' or 'rejected' responses in its reward modelling (Touvron et al., 2023). Following the preference idea, using pairwise comparisons in evaluating the LLM outputs has also been studied to some extent recently (Chen et al., 2023; Shen et al., 2023), where Wang et al. (2023b) find that pairwise comparison exhibits better human correlations than traditional scoring-based LLM evaluators. However, Zheng et al. (2024) point out the intractability of pairwise comparisons due to its quadratic query complexity (Liusie et al., 2024), which significantly limits its practicability. Qin et al. (2023) propose pairwise ranking with sorting but ignore the important transitivity of LLM predictions. In our work, we address the aforementioned issues by searching in a tractable pairwise search and transitive space based on model uncertainty.

## 5    Experiments

### 5.1   Experimental Setup

**Datasets.** We evaluate PAIRS using three commonly used meta-evaluation benchmarks for two types of NLG tasks: abstractive summarization and creative story generation. Following previous research, we adopt two evaluation strategies, namely sample-level and dataset-level Spearman correlations, as detailed in Appendix §E.1. For summarization task, we select News Room (Grusky et al., 2018) and SummEval (Fabbri et al., 2021), consistent with prior studies (Zhong et al., 2022; Fu et al., 2023). For the story generation task, we include HANNA (Chhun et al., 2022).[3] We refer to Appendix §D for more details.

**Models.** To cover a good range of LLMs, we select two open-source models: Mistral 7B (Instruct-v0.1) (Jiang et al., 2023a) and Llama-2-chat 7b (Touvron et al., 2023), and two closed-source models: GPT-3.5-turbo (-1106) and GPT-4-turbo (-preview) (OpenAI, 2023). For open-source models, we calculate the preference probability by normalizing the logits of choice 'A' and 'B'. For GPT-3.5 and GPT-4, we return top-5 log-probabilities, which can be normalized to find the preference probability.

---

[3]We make the first application of pairwise preference to evaluate over *thousands* samples. We use an anchor size of 100 on PAIRS-beam-scaled in HANNA.

| Models | News Room | | | | SummEval | | | | HANNA | | |
|---|---|---|---|---|---|---|---|---|---|---|---|
| | CH | RE | IN | FLU | CH | FLU | CON | RE | CH | SU | CX |
| **Other Metrics** | | | | | | | | | | | |
| BertScore | 0.15 | 0.16 | 0.13 | 0.17 | 0.28 | 0.19 | 0.11 | 0.31 | 0.25 | 0.22 | 0.30 |
| GPTScore (d02) | 0.31 | 0.35 | 0.26 | 0.31 | 0.28 | 0.31 | 0.38 | 0.22 | 0.22 | 0.25 | 0.08 |
| UniEval (single best) | - | - | - | - | 0.54 | 0.43 | 0.47 | 0.46 | - | - | - |
| BARTScore (CNN) | 0.65 | 0.56 | 0.61 | 0.64 | 0.45 | 0.35 | 0.38 | 0.35 | - | - | - |
| **Mistral 7B** | | | | | | | | | | | |
| Scoring | 0.32 | 0.39 | 0.20 | 0.26 | 0.23 | 0.19 | 0.37 | 0.19 | 0.30 | 0.26 | 0.37 |
| G-Eval | 0.36 | 0.36 | 0.24 | 0.39 | 0.25 | 0.20 | **0.39** | 0.25 | **0.34** | 0.25 | **0.39** |
| PAIRS-greedy | **0.57** | 0.52 | **0.53** | 0.44 | 0.25 | 0.08 | 0.28 | **0.29** | - | - | - |
| PAIRS-beam | 0.55 | **0.53** | 0.48 | **0.48** | **0.28** | **0.24** | 0.30 | 0.27 | 0.29 | **0.27** | 0.31 |
| **Llama-2-chat 7B** | | | | | | | | | | | |
| Scoring | 0.02 | 0.14 | -0.02 | 0.01 | 0.12 | 0.06 | 0.18 | 0.11 | 0.25 | 0.17 | 0.33 |
| G-Eval | 0.05 | 0.14 | -0.15 | 0.04 | 0.15 | 0.07 | 0.23 | 0.20 | **0.31** | 0.17 | **0.35** |
| PAIRS-greedy | **0.38** | 0.41 | 0.35 | 0.25 | 0.16 | 0.16 | 0.28 | 0.20 | - | - | - |
| PAIRS-beam | **0.43** | **0.43** | **0.37** | **0.28** | **0.17** | **0.18** | **0.31** | **0.24** | 0.29 | **0.19** | 0.17 |
| **GPT-3.5-turbo** | | | | | | | | | | | |
| Scoring | 0.44 | 0.47 | 0.46 | 0.50 | 0.32 | 0.28 | **0.49** | 0.25 | 0.22 | 0.12 | 0.37 |
| G-Eval | 0.45 | 0.45 | 0.45 | **0.52** | 0.30 | 0.29 | 0.47 | 0.27 | 0.28 | 0.18 | 0.39 |
| PAIRS-greedy | **0.56** | 0.55 | **0.53** | 0.45 | 0.35 | 0.23 | 0.41 | 0.27 | - | - | - |
| PAIRS-beam | **0.56** | **0.57** | 0.52 | 0.49 | **0.42** | **0.37** | 0.45 | **0.29** | 0.38 | 0.29 | 0.47 |
| **GPT-4-turbo** | | | | | | | | | | | |
| Scoring | 0.55 | 0.54 | 0.57 | **0.60** | 0.44 | 0.42 | 0.46 | 0.53 | 0.35 | 0.28 | 0.38 |
| G-Eval | 0.58 | 0.55 | 0.57 | 0.58 | 0.45 | 0.42 | 0.45 | 0.54 | 0.34 | 0.31 | 0.41 |
| PAIRS-greedy | 0.62 | 0.59 | 0.62 | 0.59 | **0.53** | 0.45 | **0.47** | 0.58 | - | - | - |
| PAIRS-beam | **0.64** | **0.61** | **0.67** | **0.60** | **0.53** | **0.48** | **0.47** | **0.59** | **0.44** | **0.36** | **0.49** |

Table 1: LLM evaluation results. We report Spearman correlations on all datasets in terms of the following evaluation aspects: coherence (CH), fluency (FLU), relevancy (RE), informativeness (IN), consistency (CON), surprise (SU), and complexity (CX). For PAIRS-beam, the experiments were conducted with a beam size of 1000 and an uncertainty threshold at 0.6.

**Baselines.** We consider two types of reference-free baselines: optimized and zero-shot evaluators. Optimized evaluators, UniEval (Zhong et al., 2022) and BARTscore (Yuan et al., 2021), mitigate the evaluation standard gaps using annotated data, making them specialized for a specific dataset or domain but not transferable. We treat them as strong but *unfair* metrics and use their reported scores as a reference. For zero-shot evaluators, we re-implement the following baselines: BERTScore (Zhang et al., 2020), G-Eval (Liu et al., 2023a), and GPTScore (Fu et al., 2023). We report direct scoring and G-Eval as the main baseline for fair comparisons.

## 5.2 Main Experiments

**Performance Gap between Direct Scoring and PAIRS.** As shown in Table 1, PAIRS, as a zero-shot general evaluator, achieves state-of-the-art (SOTA) or near-SOTA performance on all datasets, even when compared with the optimized metrics UniEval and BARTScore, which are trained on dataset-specific data. In general, PAIRS outperforms direct scoring and G-Eval baselines in most aspects, with some exceptions, such as the consistency (CON) aspect of SummEval. We attribute this to the concentrated human scores distribution, with 86.7% summaries labelled with a score of 5. These characteristics make it difficult for pairwise comparisons to yield meaningful comparisons and consequent rankings.

**Impact of PAIRS on Different (Families of) LLMs.** PAIRS particularly benefits the performance of smaller LLMs. For Mistral 7B and Llama-2 7B, we observe that pairwise comparison methods generally outperform direct scoring on News Room and SummEval. Notably, Mistral 7B achieves performance comparable to GPT-3.5-turbo on News Room. We hypothesize that this is due to their potentially similar quality of instruction tuning. Llama-2 7B struggles to effectively rate summaries with scores but demonstrates much better performance when comparing summaries. For HANNA, both small models do not benefit much from pairwise comparison, possibly due to the difficulty of understanding the long-context

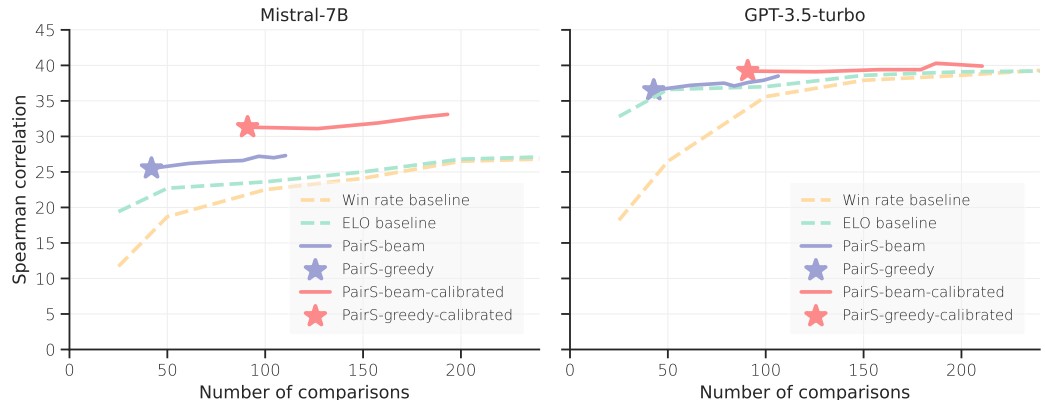

Figure 6: PAIRS-*greedy and* PAIRS-*beam are more efficient than other pairwise rank aggregation methods.* Both diagrams are evaluating on the coherence of SummEval. Each datapoint contains 16 summary candidates, with the number of pairwise comparisons ranging from 1 to 240. PAIRS-beam curves show results for beam sizes 1, 2, 5, 10, 20, 50, and 100, without uncertainty pruning. When beam size is 1, PAIRS-beam and PAIRS-greedy are equivalent.

prompts or the presence of length bias. Conversely, direct scoring performs relatively well compared to summarization tasks. We believe this is because story generation evaluation does not depend on input, making the task more straightforward. For the two closed-source GPT models, PAIRS generally outperforms direct scoring and G-Eval baselines on all three datasets, indicating that PAIRS can achieve better performance more consistently for more capable models.

**Benefits of Uncertainty-Guided Search.** PAIRS-beam generally exhibits higher correlations and lower Standard Error (SE), as shown in Table 2, than PAIRS-greedy on all datasets, indicating its ability to more robustly produce better rankings. Interestingly, the performance difference between PAIRS-greedy and PAIRS-beam is minimal for GPT-4-turbo. We attribute this to GPT-4-turbo's better transitivity, allowing the greedy search to robustly find near-optimal rankings. There are also a few cases where PAIRS-beam cannot outperform PAIRS-greedy. This could occur when the LLM's preference standards, the likelihood term, do not perfectly match with human annotations.

## 5.3 Efficiency Comparison with Rank Aggregation Baselines

**PAIRS is an efficient pairwise rank aggregation method.** We evaluate our method against two baseline rank aggregation techniques: 1) Win-loss rate and 2) ELO rating, which can both aggregate rankings from any number of pairwise comparisons. To assess their efficiency as aggregation algorithms, we calculate the correlations between human annotations and rankings predicted by PAIRS and the baselines using varying numbers of comparisons. The comparisons were random sampled without replacement. We also include the calibrated PAIRS, which requires double the computation by considering both permutations for each comparison pair.

Figure 6 demonstrates that PAIRS is more efficient than the baselines. PAIRS-greedy typically requires only about 30% of the comparisons to achieve performance similar to ELO rating. The calibrated PAIRS generally outperforms the upper bound of the baselines, where all $N(N-1)$ pairwise comparions are utilized. We attribute the strong performance of PAIRS to its leveraging of the transitivity assumption of LLMs to identify the most informative comparison pairs, which provides a natural efficiency advantage. While win-rate and ELO aggregation methods are designed to handle systems or players with variable performance, making them more effective at dealing with changes, our task involves ranking constant response candidates on a specific aspect. In this context, sorting-based algorithms that exploit transitivity have inherent advantages.

| Models | News Room | SummEval | HANNA |
|--------|-----------|----------|-------|
| M-7B greedy | 55.7±1.62 | 25.6±0.89 | 33.2±3.2 |
| M-7B beam | 54.7±0.87 | 27.6±0.92 | 33.8±2.9 |
| L-7B greedy | 39.5±1.66 | 14.9±1.94 | 29.4±2.7 |
| L-7B beam | 43.2±1.33 | 16.0±1.43 | 33.9±1.5 |
| G-3.5 greedy | 54.6±1.57 | 36.2±0.58 | 33.7±2.2 |
| G-3.5 beam | 55.9±0.72 | 41.7±0.43 | 36.2±1.0 |

| Models | News Room | SummEval | HANNA |
|--------|-----------|----------|-------|
| M-7B | 55.6 | 25.6 | 33.0 |
| - calibrated | 57.7 | 32.9 | 34.2 |
| L-7B | 38.9 | 15.6 | 29.2 |
| - calibrated | 40.5 | 16.2 | 31.5 |
| G-3.5 | 54.2 | 36.8 | 37.6 |
| - calibrated | 56.7 | 38.5 | 37.8 |

Table 2: *Quantifying transitivity.* Mean and std. of Spearman correlations (CH aspect) over 10 runs with different random seeds.

Table 3: *Calibrating* PAIRS. Mean of Spearman correlations (CH aspect) over 10 runs of PAIRS-greedy, w. and w/o calibration.

## 5.4 Ablation Studies

**Transitivity.** The transitivity assumption plays a crucial role in the effectiveness of LLMs as evaluators, impacting the speed and consistency with which they achieve rankings. Hence, it is a good measurement of the capability of LLMs as evaluators. We quantify transitivity through the analysis of correlation variations, specifically the Standard Error. We conducted 10 runs of both PAIRS-greedy and PAIRS-beam with different random seeds. For the HANNA dataset, we sample and shuffle the same subset of 200 stories. Table 2 shows that PAIRS-beam generally demonstrates better correlations and lower SE than PAIRS-greedy, which indicates that it is more robust. We notice that GPT-3.5-turbo consistently exhibits fewer errors across all datasets, suggesting that it can produce more consistent evaluations with itself, thus possessing better transitivity. Furthermore, different datasets show varying levels of errors, which we believe indicates the transitivity difficulty of each dataset. For example, compared to other datasets, HANNA has relatively higher errors, which means that, for the same base models, its optimal rankings are more difficult to find. This observation is expected as HANNA, being a creative story-writing dataset, has naturally more subjective human annotations, causing challenges for LLM alignment.

**Calibration.** Unlike direct scoring, pairwise comparison has a known preference prior, which is uniform. We perform the same calibration method as in the previous work by Zheng et al. (2023), which averages the probabilities of both permutations of the candidates. Table 3 demonstrates that calibration consistently improves the performance of Mistral 7B and Llama-2 7B. However, calibration on GPT-3.5-turbo yields relatively small improvements, likely due to its heavily skewed output probabilities (also shown in the example in Figure 7, Appendix §F.2). The over-confident predictions make the language model difficult to calibrate.

## 6 Conclusion

In this work, we have first conducted a systematic analysis of LLMs used as direct scorers for evaluation in generative tasks. The analysis has revealed a misalignment between LLM and human evaluators, which cannot be effectively mitigated by applying calibration techniques. Inspired by RLHF, we have then proposed the use of pairwise comparisons which should produce evaluation rankings that are inherently more aligned to human judgements. We have formulated the evaluation as a ranking problem from the perspective of transitivity and proposed PAIRS, an efficient, scalable, and uncertainty-guided ranking method that navigates the pairwise comparison space and estimates the MLE rankings. Extensive experimental results have demonstrated that PAIRS is more aligned with human annotations and achieves SOTA performance compared to direct scoring and domain-optimized baselines. Furthermore, we have shown that PAIRS can also be used to measure LLM's transitivity.

## Acknowledgements

The work is supported by the UK Research and Innovation (UKRI) Frontier Research Grant EP/Y031350/1 (the UK government's funding guarantee for ERC Advanced Grants) awarded to Anna Korhonen at the University of Cambridge. The work has been supported in part by a Royal Society University Research Fellowship (no 221137; 2022-) awarded to Ivan Vulić, and by the UK EPSRC grant EP/T02450X/1.

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

## A  Additional Related Work

**Calibrating LLM Evaluators.** Though previous studies have shed light on using LLMs as evaluators, LLMs have been exposed to their sensitivity to prompt designs (Zhao et al., 2021; Zhou et al., 2023), and even biased from various factors, including but not limited to positional bias (Wang et al., 2023b; Pezeshkpour & Hruschka, 2023), verbosity bias (Saito et al., 2023), and contextual bias (Zhou et al., 2024b). Calibration methods are then proposed to mitigate these biases by criteria optimization (Liu et al., 2023b), context decomposition (Li et al., 2023b), order permutation (Wang et al., 2023b; Zheng et al., 2023), ensembling (Li et al., 2023a) and batching (Zhou et al., 2024b). However, we show that calibrations are yet insufficient in aligning LLM evaluators with humans.

## B  Future Work

By formulating the evaluation as a ranking problem, we provide a playground for future work to develop **better searching algorithms**. PAIRS is a type of Breadth-First Search (BFS) method. However, under different transitivity assumptions, there may be a better trade-off between performance and efficiency. We encourage future work to explore and develop novel ranking algorithms that can optimize this trade-off.

Furthermore, future work can investigate **alternative scaling approaches**. For instance, by examining the performance of LLMs on selecting the best candidate from a set of K options, instead of performing pairwise ranking, the search efficiency could potentially be improved from $O(log_2 N)$ to $O(log_K N)$. The improvement in efficiency could significantly impact the scalability and practicality of the search-based method, especially when dealing with large-scale datasets or real-world applications.

## C  Derivations

### C.1  Rank Aggregation Likelihood Derivation

In the general setting of **learning to rank**, we have a set of items and a set of pairwise comparisons between these items. Each pairwise comparison indicates which of the two items is preferred or ranked higher. The goal is to find a ranking of the items that is most consistent with these pairwise comparisons.

The likelihood function for the general ranking problem can be written as:

$$\mathcal{L}(\pi) = \prod_{(i,j) \in C} P(i \succ j | \pi).$$

Here, $C$ is the set of all pairwise comparisons, and $P(i \succ j | \pi)$ is the probability of observing the comparison 'item i is preferred over item j' given the ranking $\pi$.

In the **rank aggregation** problem, we assume that the pairwise comparisons are independent and that the probability of observing a comparison 'item i is preferred over item j' is given by $P(i \succ j)$, which is provided as input. Therefore, the likelihood function for the rank aggregation problem can be derived from the general likelihood function by making the following substitution:

$$P(i \succ j | \pi) = \begin{cases} P(i \succ j) & \text{if } \pi(i) < \pi(j) \\ 1 - P(i \succ j) & \text{if } \pi(i) > \pi(j). \end{cases}$$

In other words, if the ranking $\pi$ is consistent with the comparison 'item i is preferred over item j', then the probability of observing this comparison is simply $P(i \succ j)$. Otherwise, if the ranking $\pi$ is inconsistent with the comparison, the probability is $1 - P(i \succ j)$.

Substituting this into the general likelihood function, we get:

$$\mathcal{L}(\pi) = \prod_{i,j:\pi(i)<\pi(j)} P(i \succ j) \times \prod_{i,j:\pi(i)>\pi(j)} (1 - P(i \succ j)).$$

The second product can be simplified by noting that $P(j \succ i) = 1 - P(i \succ j)$, so we get:

$$\mathcal{L}(\pi) = \prod_{i,j:\pi(i)<\pi(j)} P(i \succ j) \times \prod_{i,j:\pi(i)>\pi(j)} P(j \succ i).$$

Finally, since the comparisons are assumed to be independent, the second product can be absorbed into the first one, giving us the likelihood function for the rank aggregation problem:

$$\mathcal{L}(\pi) = \prod_{i,j:\pi(i)<\pi(j)} P(i \succ j).$$

### C.2 Transitive Ranking Likelihood Derivation

In this section, we will show how we derive the transitive ranking likelihood, Equation (4), from the non-transitive ranking likelihood, Equation (3), given the compositional stochastic transitivity assumption.

Compositional stochastic transitivity (Fishburn, 1973; Latta, 1979; Oliveira et al., 2018) states that $P(a \succ c) = H(P(a \succ b), P(b \succ c))$, where $H : \mathbb{R} \to [0, 1]$ is some strictly-monotone and symmetric composition rules. We use the notation of $P_{a,b}$ to represent $P(a \succ b)$. Therefore, we can have $P_{a,c} = \lambda P_{a,b} + \lambda P_{b,c}$, where $\lambda \in [0, 0.5]$.

For a ranking $\pi_{1:N}$, where $\pi_1 \succ \pi_2 \succ ... \succ \pi_N$, of length $N$, there are $N - k$ pairwise comparisons with distance $k$, e.g. $P_{\pi_i, \pi_{i+k}}$. Each comparison probability with distance $k$ can be expressed by comparison probabilities with distance 1 using $k - 1$ coefficients:

$$\begin{aligned}
P_{\pi_i, \pi_{i+k}} &= \lambda_1 P_{\pi_i, \pi_{i+1}} + \lambda_1 P_{\pi_{i+1}, \pi_{i+k}} \\
&= \lambda_1 P_{\pi_i, \pi_{i+1}} + \lambda_2 P_{\pi_i, \pi_{i+2}} + ... + \lambda_{k-1} P_{\pi_{i+k-2}, \pi_{i+k-1}} + \lambda_{k-1} P_{\pi_{i+k-1}, \pi_{i+k}}.
\end{aligned} \tag{6}$$

The number of $\lambda$ coefficients has a scale of $O(N^3)$. The non-transitive ranking likelihood can be expressed by probabilities of comparisons with distance 1.

$$\mathcal{L}_{NT}(\pi) = p(\pi_{1:N}) = \mu_1 P_{\pi_1, \pi_2} + \mu_2 P_{\pi_2, \pi_3} + ... \mu_{N-1} P(\pi_{N-1}, \pi_N) \tag{7}$$

If letting $\mu_1 = \mu_2 = ... = \mu_{N-1}$, we can have around $O(N^2)$ constraint equations, expressed by $\lambda$s. As the dimension of $\lambda$ coefficients is much larger than the number of constraint equations, we argue that there should always be a set of $\lambda \in [0, 0.5]$ satisfying the constraints.

### C.3 Ranking to Scores Conversion via Quantile Matching

Given ranking information, we can convert it to scores with any desired score distribution, via Quantile or Cumulative Density Function (CDF) matching. The desired score distribution can be either an assumed human score prior, for example, a normal distribution is a simple and suitable guess on prior, or an estimated distribution from validation samples.

To match the CDF of variable $Y$ to variable $X$, we need to find a transformation function $g$ such that $Y = g(X)$. The goal is to make the CDF of $Y$ equal to the CDF of $X$.

The CDF matching transformation is given by:

$$Y = F_Y^{-1}(F_X(X)), \tag{8}$$

where $F_Y^{-1}$ is the inverse of the CDF of $Y$.

For example, if we assume the score prior distribution is $[10\%, 20\%, 40\%, 20\%, 10\%]$ for scores 1 to 5, then we can simply convert a ranking into scores by assigning the smallest 10% with score 1, the smallest $10\% - 30\%$ with score 2 and so on.

## D Datasets

**Summarization Tasks**: We select SummEval (Fabbri et al., 2021) and NewsRoom (Grusky et al., 2018) for our experiments. SummEval consists of 100 input source texts, each paired

with 16 summary candidates generated by different LMs. The dataset is annotated on four aspects: coherency (CH), fluency (FLU), consistency (CON), and relevancy (RE). NewsRoom includes 60 input source texts and 7 output summaries for each sample. It is annotated on the aspects of coherence (CH), fluency (FLU), relevance (RE), and informativeness (IN).

**Story Generation Task**: We select HANNA (Chhun et al., 2022), which contains 1056 creative story writings generated from 96 prompts collected from WritingPrompt. Three representative aspects are selected: coherence (CH), surprise (SU), and complexity (CX). We concatenate the story prompts with the story bodies, allowing the stories to be compared without input prompts. Due to the computational cost, we use the scaling variant of PAIRS with an anchor size of 100.

# E  Implementation Details

## E.1  Evaluation Strategies

Following previous works (Liu et al., 2023a; Zhong et al., 2022; Fu et al., 2023) we used two evaluation strategies, Sample level and Dataset level. For the sample level, the correlation values are calculated across the multiple candidates to the same input and then averaged across all input samples. For the dataset level, the correlation is calculated across the entire dataset.

# F  Additional Experiments

## F.1  Scaling Variant Performance Loss

| Methods | Mistral-7B | | Llama2-7B | |
|---|---|---|---|---|
| | $\rho$ | # Q | $\rho$ | # Q |
| greedy | 30.4 | 9,762 | 30.4 | 9,065 |
| beam | 32.0 | 24,733 | 31.1 | 24,006 |
| beam-scaled | 29.3 | 7,571 | 29.5 | 7,438 |

Table 4: *Performance trade-off of the scaling variant*. We compare the Spearman correlations ($\rho$) and numbers of the model query (# Q) for PAIRS-greedy, PAIRS-beam and PAIRS-beam-scaled on the Coherence aspect of the HANNA dataset. The beam size is set to 100, the anchor size is set to 100, and the uncertainty threshold is set to 0.67.

Because of the binary search in the second stage of the scaling variant, we expect there will be performance loss compared to PAIRS-greedy and PAIRS-beam. To investigate the actual trade-off, we conduct an ablation study using Mistral-7B and Llama2-7B. The results presented in Table 4 illustrate the trade-off between performance and computational cost. PAIRS-beam-scaled achieves comparable performance to PAIRS-greedy, while requiring fewer model queries. We note that the computational complexity of PAIRS-greedy grows at a rate of $O(N \log N)$, whereas PAIRS-beam-scaled will grow at a rate of $O(\log 100)$, making it more computationally efficient as the problem size increases.

## F.2  Uncertainty Threshold for Different LLMs

Different base models have different logit distributions. Mistral 7B and Llama2 7B tend to have symmetric and bell-shaped distributions, which means a large portion of the probabilities lie within the range of $[0.3, 0.7]$, as shown in Figure 4. However, as shown in Figure 7, the probability distribution is different for GPT-3.5-turbo. Therefore, their optimal uncertainty thresholds would be different.

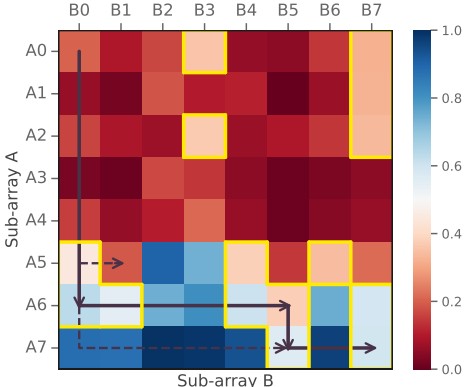

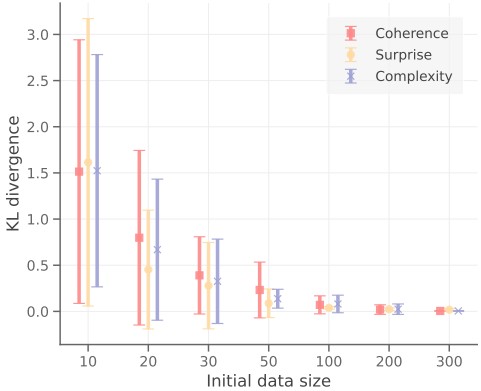

Figure 7: Illustration of PAIRS with a beam size of 2. Heatmap values show $P(A_i \succ B_j)$ by GPT-3.5-turbo. The rest of the settings are the same as Figure 4

Figure 8: Empirical evidence for anchor set size selection. The error bars show the mean and standard deviation of the KL divergence between the score distributions of randomly sampled anchor sets (100 repetitions) and the full set. The results suggest that an anchor set size of 100 yields a relatively small error.

### F.3 Theoretical and Empirical Evidence for Choice of Anchor Set Size

The Likert scale score rating can be treated as a five-category classification problem. Therefore the problem of anchor size estimation becomes estimating the sample size for estimating a categorical distribution. This can be calculated by the method of 'proportion sample size calculation' The formula for the sample size calculation for a single proportion is:

$$n = \frac{(z^2 \times p \times (1 - p))}{e^2},$$

where $n$ is the sample size, $z$ is the z-score corresponding to the desired confidence level, $p$ is the expected proportion of the most common category, and $e$ is the desired margin of error. To account for multiple categories, the sample size is multiplied by a design effect ($deff$), which typically ranges from 1.5 to 2.

$$n_{adj} = n \times deff,$$

where $n_{adj}$ is the adjusted sample size for multiple categories, $n$ is the sample size calculated for a single proportion, and $deff$ is the design effect. The design effect accounts for the increased variability in the data due to the multiple categories and the potential clustering of responses within categories.

If we take the desired confidence level as 0.8, the expected proportion of the most common category $p$ as 0.4, error margin $e$ as 7% and $deff$ as 1.5, then the sample size required for estimating five categories distribution is around 120.

Figure 8 presents **empirical evidence** for selecting an appropriate anchor set size. To determine the optimal size, we conducted an experiment where we randomly sampled different numbers of annotations from the HANNA (coherence) dataset and calculated the Kullback-Leibler (KL) divergence between the score distributions of the sampled sets and the entire set. For each anchor set size, we repeated the sampling process 100 times and computed the standard errors. The results suggest that an anchor set size between 100 and 200 provides a reasonable balance between computational efficiency and representativeness of the full dataset, as the KL divergence stabilizes within this range.

## G  Prompt Templates

### G.1  Prompts

---

Prompt Template Example for Pairwise Comparison

---

```
Evaluate and compare the coherence of the two following summaries.

Source text: [source_text_input]

Summary A: [summary_1]

Summary B: [summary_2]

Question: Which summary is more coherent?
If the summary A is more coherent, please return 'A'.
If the summary B is more coherent, please return 'B'.
You must only return the choice.
Answer: [output]
```

---

Table 5: A prompt template example for pairwise comparison of the **coherence** aspect in the **summarization** task.

---

Prompt Template Example for Direct Scoring

---

```
Evaluate the coherence of the following summary.

Source text: [source_text_input]

Summary: [summary]

Please rate on a scale from 1 to 5, where 1 represents very low coherence
and 5 indicates excellent coherence. You must only return an int score.
Score: [output]
```

---

Table 6: A prompt template example for direct scoring of the **coherence** aspect in the **summarization** task.

