# OpenReview forum: "Aligning with Human Judgement: The Role of Pairwise Preference in Large Language Model Evaluators"
_colmweb.org/COLM/2024/Conference — COLM_

### Official Review · Reviewer_6K4e · 2024-05-09

**Rating:** 7
**Confidence:** 3
**Ethics Flag:** 1

**Summary:**

This paper develops a method for automatically ranking a set of text responses (e.g., summaries of source texts) using LMs, while training to maintain alignment with human rankings.

Problem background: Simply prompting an LM to score texts according to criteria can result in significant misalignment with human judgements. Estimating the marginal human score distribution p_H(s) and using that to calibrate the LM scores does not fully resolve the issue, implying that the model's likelihood p_M(y|s) is also misaligned from the human likelihood.

Method: the proposed method (PAIRS) instead focuses on using pairwise comparisons between candidate texts to produce a global ranking. This is formulated as searching over an ordering (permutation) that maximizes the likelihood; the definition in Eq 4 is designed to considers two algorithms for finding this ordering. The first is a merge-sort-like algorithm that merges two sorted lists of responses by greedily choosing the next candidate from either list that locally maximizes the likelihood. They then introduce a beam-search-like alternative of this procedure that tracks the top-k beams of candidate rankings at any time.

The experiments show that PAIRS achieves strong correlation with gold annotations on several natural language generation benchmarks.

**Questions To Authors:**

* What is the beam size used for PairS-beam(-scaled) in Figure 5? Does that matter at all for the number of model calls?
* If I understand correctly, the scaling variant of Pairs is linear in number of samples to be ranked, is that asymptotically the same as direct scoring?

**Reasons To Accept:**

The proposed method is novel to my knowledge, seems relatively straightforward to implement, and the experiments appear to demonstrate strong empirical results.

**Reasons To Reject:**

* The text in 3.2 is not very clear at explaining how PairS-beam works, and the figure/algorithm need to be carefully studied to infer how it works without much supporting text. The caption or text could be expanded to make this easier to understand.
* The results could benefit from more clarity about the actual real-world cost (in terms of # of model calls) of each method in the experiments. Do the proposed methods use more model calls than direct scoring?

---

> ### Author Rebuttal · Authors · 2024-05-30
>
> We thank the reviewer for their insightful feedback while acknowledging the novelty, effectiveness and simplicity of our method. We appreciate your valuable suggestions in further discussing the real-world cost. With the additional results with respect to beam size and model calls, we hope that our response sufficiently addresses the reviewer’s concerns and the reviewer could consider improving their score.
>
> ---
> > Presentation in Section 3.2
>
> We sincerely appreciate the time and effort you have invested in carefully examining our algorithm and the corresponding demo presented in Algo.1, Fig.3 and 4. We acknowledge that our method is intricate and our presentation may not have been optimally structured to facilitate a clear understanding.
> We will expand the text and captions to improve the clarity of our work.
>
> ---
> > Beam size, number of model calls and performance
>
> We agree that detailed information on the number of model calls can help readers better understand the complexity of our methods and baselines.
> Number of model calls is affected by many factors, including beam size, uncertainty threshold, number of candidates to rank and the distribution of model’s output logits.
> Therefore, we conducted an additional ablation study on beam size selection, keeping the uncertainty threshold at 0.6. We include the Spearman correlation results along with the number of model calls in ():
>
> |Beam Size |Mistral-7B|Llama2-7B|
> |-|-|-|
> |1|22.9 (39.3)|13.5 (34.2)|
> |5|23.5 (73.7)|15.1 (70.5)|
> |20|23.8 (91.1)|16.6 (91.5)|
> |100|25.3 (103.8)|16.8 (94.6)|
>
> As mentioned in Section 3.4, direct scoring methods generally have complexity of O(N), while existing pairwise baselines are generally O(N^2). PairS provides an O(NlogN) solution and a trade-off between performance and complexity between O(NlogN) and O(N^2).
>
> In conclusion, PairS is more efficient than pairwise baselines and utilizes computation more effectively than some score-based evaluation methods. For instance, G-Eval uses a weighted average of 20 inference repetitions, resulting in a total computation of 20N. In comparison, PairS-greedy requires Nlog2(N) in the worst case. Specifically, for the 16-candidate SummEval example mentioned above, the factor is only 2.5N.
>
> ---
> > Asymptotic complexity of PairS scaling variant:
>
> In the second stage of PairS-beam-scaled, the complexity is N*log2(Anchor size). With an anchor size of 200, the complexity is 7.6N, which is comparable to some direct scoring method such as G-Eval.

---

> > ### Author Response · Authors · 2024-06-05
> > **Reminder**
> >
> > Dear Reviewer 6K4e,
> >
> > We thank you once again for your feedback! As the discussion period is ending, we wonder if you could take a look at our response if you haven't? We are eager to know whether our response has addressed your concerns: we'd be grateful if you could improve the rating if that's the case. Otherwise, we are always happy to answer further questions that you may have.
> >
> > Best,
> >
> > Authors

---

> > > ### Comment · Reviewer_6K4e · 2024-06-05
> > >
> > > Thank you to the authors for their response, the additional ablation on beam size is helpful. I feel that my questions have largely been clarified and have no further questions.

---

### Official Review · Reviewer_VVGt · 2024-05-11

**Rating:** 9
**Confidence:** 4
**Ethics Flag:** 1

**Summary:**

This paper presents an LLM evaluation framework with pairwise comparisons (Qin et al. 2023). Pairwise comparisons require $O(N^2)$ evaluations where $N$ is the number of outputs from LLMs. So, this paper explores the use of transitivity of preferences to reduce the number of evaluations into $O(N \log N)$ and presents a sorting algorithm for LLM outputs, inspired by the merge sort algorithm. This study also proposes using beam search for merge operations because transitivity may not always hold. The experimental results demonstrate that the proposed method has higher correlations with human annotations and performs better than direct scoring and domain-optimized baselines.

**Questions To Authors:**

The title includes "aligning with human judgment." I was expecting to see how the proposed method aligns LLM judgments with humans. However, this study does not directly align human judgment with that of an LLM evaluator. In fact, this paper gave up calibrating LLM evaluators directly for alignment, but obtained a better alignment as a result of utilizing pairwise comparisons. So I felt that this phrase in the title is a bit misleading. Is it right that evaluator LLMs were not trained to align human pairwise preferences?

Where can I find the numbers of pairwise comparisons of the PAIRS-greedy, PAIRS-beam, and Qin et al. (2023)?

**Reasons To Accept:**

+ This paper is well-structured and written.
+ This paper proposes a reasonable algorithm for making pairwise approaches feasible for LLM comparisons.
+ The experiments demonstrate the superiority of the proposed approach, which has a great impact on LLM evaluation and development.

**Reasons To Reject:**

None

---

> ### Author Rebuttal · Authors · 2024-05-30
>
> We thank the reviewer for their valuable feedback and for recognizing the well structured presentation, novelty and superiority of this work to the research community. We appreciate the suggestions for listing the number of pairwise comparisons in our work to further enhance the presentation. Following this additional experiment, we hope our response could sufficiently address the reviewer’s concerns.
>
> ---
> > A better title of "aligning with human judgment."
>
> Thank you for your suggestions. We are aware that the current title may overlap with the line of work in fine-tuning LLMs or calibration for a better alignment with human values. Our original intent was to highlight PairS' ability to leverage a smaller gap in evaluation standards between humans and LLMs. We will endeavour to figure out a more appropriate title for future versions.
>
> ---
> > Number of pairwise comparisons
>
> Thank you for the suggestion to report the actual number of model calls. We agree the actual number of model calls can be more helpful than the computational complexity. We conducted an additional ablation study on beam size selection, keeping the uncertainty threshold at 0.6. We will include the Spearman correlation results along with the number of model calls in ():
>
> |Beam Size |Mistral-7B|Llama2-7B|
> |-|-|-|
> |1|22.9 (39.3)|13.5 (34.2)|
> |5|23.5 (73.7)|15.1 (70.5)|
> |20|23.8 (91.1)|16.6 (91.5)|
> |100|25.3 (103.8)|16.8 (94.6)|
>
> We will include this ablation study with a detailed figure in the later version.

---

> > ### Comment · Reviewer_VVGt · 2024-06-05
> > **Re: Rebuttal by Authors**
> >
> > Thank you for the response. It's my pleasure to contribute to the work in some way through the reviewing process.

---

> > > ### Author Response · Authors · 2024-06-05
> > > **Thanks**
> > >
> > > Thank you for your kind message. We truly appreciate the time and effort you have put into reviewing our work. Thank you for your constructive feedback and for being a part of this collaborative effort to advance our field of study.
> > >
> > > Best regards,
> > >
> > > Authors

---

### Official Review · Reviewer_zw92 · 2024-05-13

**Rating:** 4
**Confidence:** 4
**Ethics Flag:** 1

**Summary:**

This paper develops a search-based search algorithm called PAIRS, which searches example pairs for LLM pairwise judgment to create a global ranking of examples. The basic idea is to convert the pairwise judgments conducted so far to prune the pairs for which the judgements can be inferred. Experiment results show that PAIRS can produce judgments that are more correlated with human judgments than a scoring-based method G-Eval.

**Questions To Authors:**

Please respond to the concerns raised in Reasons to Reject.

**Reasons To Accept:**

- (S1) Aggregating pairwise LLM judgments into a global ranking is an interesting idea and not well studied. LLM-as-a-Judge is an active research area and the paper could contribute to the research community.
- (S2) The paper is generally well-written.

**Reasons To Reject:**

- (W1) Baseline methods are missing. It is not clear if the proposed solution is reasonably effective for the purpose. More specifically, on (1) search and (2) aggregation. For (1), the most naive way is to use randomly selected pairs with the same rank aggregation method. For (2), other aggregation methods such as the Elo score, which is commonly used for LLM ranking (eg., Chatbor Arena).
- (W2) The motivation for Evaluation as Ranking is unclear. The main purpose of evaluation in most cases is to compare two (or multiple systems) to judge which one is better (best). In that sense, the global ranking discussed in the paper may not be the best option for evaluation.For example, the SummEval dataset used to 1-5 ratings for manual evaluation and the granularity is much more fine-grained by the proposed method.
- (W3) The cost of the proposed method is expensive. The computational complexity for scoring-based methods such as G-Eval is O(N). It is not clear if it’s worth paying extra cost. One option is to conduct experiments under the same budget (ie., just conducting N pairs to be fair against scoring-based methods) and discuss if the pairwise judgment approach can be more effective.
- (W4) Section 2 is misleading, as the limitations discussed in the section are not  addressed by aggregating pairwise judgments. If it is, the paper needs to show that the rating distribution is better aligned with that of human judgments (which is not straightforward, as the judgment is global ranking)

---

> ### Author Rebuttal · Authors · 2024-05-30
>
> We thank the reviewer for insightful reviews and the acknowledgement of the novelty and presentation of our work.
>
> ---
> > Pairwise Baselines
>
> We agree that pairwise baselines can better demonstrate our method's efficiency. We propose comparing PairS with two baselines: 1) Win-loss rate and 2) ELO rating, both using randomly sampled pairs without replacement.
> We present the mean and std of Spearman corr. over 10 repetitions, vs # of comparisons
> |||50|100|240(max)|
> |-|-|-|-|-|
> |Mistral-7B|Win-rate|18.7|22.5|26.8|
> ||ELO|22.7|23.6|27.1|
> ||PairS-G|23.0(39.2)|36.5(41.5*2)||
> |GPT-3.5|Win-rate|27.5|36.6|40.3|
> ||ELO|35.6|38.0|40.2|
> ||PairS-G|39.3(44.3)|39.9(46.3*2)||
>
> Uncalibrated PairS-greedy achieves the same performance as the baselines using **only 34% and 31% of the comparisons**. Calibrated PairS-greedy, which doubles the number of comparisons, can sometimes outperform baselines that use O(N^2) comparisons.
>
> ---
> >Ranking as evaluation is not motivated
>
> We respectfully disagree.
> Ranking offers a much finer granularity compared to score rating, as the number of candidates is typically much larger than the number of score categories. Even in rare cases where the candidate number is smaller, ranking still provides valuable information. As discussed in Sec 3.1, scoring suffers from challenges in aligning evaluation standards across raters.
>
> ---
> >PairS may be expensive
>
> We acknowledge that pairwise evaluations are generally more expensive than pointwise methods. However, we would like to highlight:
> 1) Performance should be considered alongside cost. We show that PairS outperforms most pointwise methods, such as G-Eval, which only obtain marginal gains with increased computation.
> 2) PairS is highly efficient compared to pairwise baselines.
>
> ---
> >The limitation in Section 2
>
> In Section 2,  we show that calibration cannot effectively bridge the gap between predictive posterior distributions. This serves as empirical evidence that leads to the conclusion that there is yet a gap between the likelihood terms of Eq.1., which motivates us to investigate pairwise evaluation to mitigate the gap in evaluation standards.
>
> It is important to note that this paper does not focus on calibration, and we are not showing that our method can better eliminate the bias in **model’s prior**. Instead, our experiments are designed to show that we achieve better final evaluation performance, which can be interpreted as a smaller gap between the **posterior terms**.

---

> > ### Author Response · Authors · 2024-06-04
> > **Following up discussion towards the rebuttal**
> >
> > We thank the reviewer again for the valuable suggestions. We hope our complementary experiments have addressed all the reviewer’s concerns. We hope that in light of the response, the reviewer could consider improving their score.
> > We also provide some further discussion to support our rebuttal
> >
> > ---
> > > Pairwise Baselines
> >
> > PairS leverages the transitivity assumption of LLMs, which gives it a natural efficiency advantage. Win-rate and ELO aggregation methods are designed to handle systems or players with non-constant performance, making them more effective at dealing with changes. However, in our case, we are ranking the exact same aspect of constant response candidates, which means sorting-based algorithms that exploit transitivity have inherent benefits.
> >
> > ---
> > > Ranking as evaluation is not motivated
> >
> > To support our motivation claim in Sec 3.1, we conducted a complementary experiment using 50 randomly sampled summary candidate pairs for each source text in SummEval. We calculated the 3-way agreement rate accuracy between direct scoring and human scores, as well as between pairwise comparisons and human scores. The results consistently show that pairwise comparisons have better agreement with human judgements than scoring, across all three selected LLMs.
> > |Model|Scoring|Pairwise|
> > |-|-|-|
> > |Mistral-7B|38.7|45.5|
> > |Llama2-7B|39.6|43.0|
> > |GPT-3.5| 42.3|48.1|
> >
> > Moreover, ranking-based evaluation has already been successfully employed in various real-world applications, such as re-rankers for data selection, output reordering, and recommendation systems.
> >
> > ---
> > > PairS may be expensive
> >
> > More discussion about the cost of pointwise methods, such as G-Eval.
> > G-Eval uses a weighted average of 20 inference repetitions, resulting in total computations of 20N. In comparison, PairS-greedy requires Nlog2(N) in the worst case. Specifically, for the 16-candidate SummEval example mentioned above, the factor is only 2.5N.

---

> > > ### Comment · Reviewer_zw92 · 2024-06-07
> > >
> > > Thank you for your responses and for sharing additional experiments. Apologies for the delayed reply.
> > >
> > > Here are my follow-up questions for each of the points in your both responses.
> > >
> > > >> Pairwise Baselines
> > > >
> > > > Uncalibrated PairS-greedy achieves the same performance as the baselines using only 34% and 31% of the comparisons. Calibrated PairS-greedy, which doubles the number of comparisons, can sometimes outperform baselines that use O(N^2) comparisons.
> > >
> > > Thank you for sharing the additional experiment results and the observations.
> > >
> > > I cannot draw the conclusion from the table. Could you add an explanation for the table in a similar style as the paper? What are these numbers? What are 50, 100, 240(max)?
> > > What do 23.0(39.2) and 36.5(41.5*2) mean for PairS-G?
> > >
> > > >> Ranking as evaluation is not motivated
> > > >
> > > > as the number of candidates is typically much larger than the number of score categories
> > >
> > > If you compared M systems for N pairs, wouldn’t the computational complexity be O(M^2 * N^2)? Can N pairs be sampled from M systems and can offer the same level of ranking quality as that for 2 systems? I think this point is not discussed or clarified in the paper.
> > >
> > > >> PairS may be expensive
> > > >
> > > > We acknowledge that pairwise evaluations are generally more expensive than pointwise methods. However, we would like to highlight:
> > > >
> > > > 1. Performance should be considered alongside cost. We show that PairS outperforms most pointwise methods, such as G-Eval, which only obtain marginal gains with increased computation.
> > > > 2. PairS is highly efficient compared to pairwise baselines.
> > >
> > > > G-Eval uses a weighted average of 20 inference repetitions
> > > Thank you for sharing the additional experiment.
> > >
> > > The G-Eval paper called the API multiple times because the GPT API does not return token probababilities. The method uses the weighted average of tokens (e.g., 1-5) based on their probabilities, which can be computed by a single inference. Below is the description from the G-Eval paper.
> > >
> > > ```
> > > For GPT-4, as it does not support the output of token probabilities, we set ‘n = 20, temperature = 1, top p = 1’ to sample 20 times to estimate the token probabilities.
> > > ```
> > >
> > > >> The limitation in Section 2
> > > >
> > > > It is important to note that this paper does not focus on calibration,
> > >
> > > Thanks for clarifying. Then, I believe Section 2 is misleading and the point should be further clarified. This is rather a presentation issue and I don't have any follow-up questions.

---

> > ### Author Response · Authors · 2024-06-05
> > **Reminder**
> >
> > Dear Reviewer zw92,
> >
> > We thank you once again for your feedback! As the discussion period is ending, we wonder if you could take a look at our response if you haven't? We are eager to know whether our response has addressed your concerns: we'd be grateful if you could reconsider the rating if that's the case. Otherwise, we are always happy to answer further questions that you may have.
> >
> > Best,
> >
> > Authors

---

> > ### Author Response · Authors · 2024-06-06
> > **Reminder**
> >
> > Dear Reviewer zw92,
> >
> > As the discussion period is ending, we would like to remind you to take a look at our response if you haven't. We are eager to know whether our response has addressed your concerns: we'd be grateful if you could reconsider the rating if that's the case. Otherwise, we are always happy to answer further questions that you may have.
> >
> > Best,
> >
> > Authors

---

> ### Author Response · Authors · 2024-06-07
> **Rebuttal 2**
>
> We thank the reviewer for the feedback and have addressed them below.
>
>
> 1. Pairwise Baselines
>
> This ablation study with other two pairwise baselines using a different format than our result Table 1. In this table, 50, 100, 240(max) represents the number of randomly selected candidate pairs used to calculate the win rate or the ELO rankings. For SummEval dataset, there are 16 candidates for each datapoint, resulting in a total of 240 pairs. (N^2, including swapped comparisons and excluding self-comparison.)
>
> For PairS-Greedy, we cannot control the exact number of comparisons needed for the ranking. By averaging the results over 10 repetitions, we use 39.2 and 44.3 comparisons to achieve the global rankings for Mistral-7B and GPT-3.5.
> When using calibrated PairS-Greedy, (where each comparison uses the average probabilities of pairwise comparison with permutated positions), the number of comparisons are 41.5*2 and 46.3*2. The reason we mention calibrated result is that the 240 total comparison pairs include comparisons with swapped positions.
>
> Both uncalibrated and calibrated PairS-Greedy, our method are significantly more efficient than the win rate and ELO baselines. Presenting these results in a table may not be optimal, and we plan to use line figures in the formal revision to better illustrate the findings.
>
> ---
> 2. Ranking as evaluation is not motivated
>
> > If you compared M systems for N pairs, wouldn’t the computational complexity be O(M^2 * N^2)?
>
> We believe the reviewer has misunderstood the setup of pairwise ranking. **We are not comparing the whole LLM systems, but a list of candidates.** The candidates can be generated by different systems, but we only evaluate the quality of those candidates. If we have M candidates to rank, we can sample up to M*2-M pairs. What we meant by "as the number of candidates is typically much larger than the number of score categories", is M is usually larger than the scale of score categories (5 for likert scale). Therefore, providing the ranking of M candidates is more fine-grain than scoring.
>
> ---
> 3. pointwise evaluation and G-Eval
>
> We acknowledge the statement in G-Eval paper. We believe the reason why G-Eval cannot handle API model efficiently is beyond the discussion of our paper. In our paper, what we claim is pointwise evaluation methods generally have complexity of O(N) and our argument is still hold: pointwise evaluations can only gain marginal performance with more computations.
>
> ---
> In conclusion, we believe our experiments and ablation studies provide a comprehensive evaluation of the proposed PairS method. We have compared with multiple baselines including both pointwise and pairwise evaluation methods. Our results show that PairS is more efficient and effective than pairwise baselines and achieves higher human agreement than both pointwise and pairwise baselines.
>
> Furthermore, we have clarified and provided explicit support for the motivation behind ranking evaluation. We also have address reviewer's misunderstanding regarding our discussion of the limitation of direct scoring in Section 2.
>
> We have made a sincere effort to respond to the reviewer's concerns and to provide additional evidence and clarifications to support our claims. In light of our response, we respectfully request that the reviewer consider improving their score, as we believe our work makes a valuable contribution to the field of LLM evaluator.

---

### Official Review · Reviewer_VjHC · 2024-05-24

**Rating:** 7
**Confidence:** 4
**Ethics Flag:** 1

**Summary:**

- This paper proposes a pairwise-preference search method that estimates MLE of preference rankings by searching through an uncertainty-pruned pairwise comparison space.
- The proposed method is more effective than traditional direct scoring evaluations and existing pairwise baselines.

**Questions To Authors:**

- Any reason for the beam size of 1000, which seems to be very large number?
- What if the beam size is smaller for the main experiments? Also, what about the correlation between the inference time and the different beam sizes?

**Reasons To Accept:**

- The writing is clear to understand.
- The analysis of the limitations of calibration in aligning LLM evaluators is well presented and the proposed PAIRS method is also well described.
- Experiments are achieved with open-source and close-source models on multiple datasets.
- Ablation studies are conducted.

**Reasons To Reject:**

- The main LLM evaluation results with the proposed method are not always better than the direct scoring method, which makes me doubt about the motivation in 3.1.
- Some results in Table 1 are close to the baselines. Are those statistically significant?
- Experiments can be more comprehensive if different size of open-source models (e.g. 70B size models). Also, it is curious if the larger models perform similarly.

---

> ### Author Rebuttal · Authors · 2024-05-29
>
> We thank the reviewer for their insightful feedback while acknowledging the quality of our work. We hope that in light of the response that addresses all the reviewer’s concerns, the reviewer could consider improving their score.
>
> ---
> >PairS performance vs Direct scoring, and the motivation in 3.1.
>
> While our main results in Table 1 show that **PairS generally outperforms direct scoring**, we acknowledge that there are a few exceptions. These variations are expected due to challenges in LLMs' capability to handle long-context input and their positional bias.
> The exceptions occur in the consistency aspect of SummEval, as explained in Section 5.2,  which is caused by the highly concentrated distribution of human scores, and in the performance of Llama2-7B and Mistral-7B on HANNA, which we attribute to the difficulty in understanding long-form creative writing.
>
> To further support our motivation stated in Section 3.1, we conducted a complementary experiment by randomly sampling 50 summary candidate pairs for each source text in SummEval and calculating the 3-way *agreement rate accuracy* between direct scoring vs. human scores and pairwise comparison vs. human scores. The results show that pairwise comparisons have *consistently* better agreement than scoring.
> |Model|Scoring|Pairwise|
> |-|-|-|
> |Mistral-7B|38.7|45.5|
> |Llama2-7B|39.6|43.0|
> |GPT-3.5| 42.3|48.1|
>
> ---
> > Significance test on Table 1 results:
>
> Thank you for raising this point. To address this, we conducted z-tests on 30 repetitions for cases where direct scoring and PairS have relatively similar performance. The results show that PairS-greedy significantly outperforms direct scoring for Mistral-7B on SummEval coherence (p<0.01). For GPT-4, the differences are significant on fluency (p<0.001) and weakly significant on consistency (p<0.15). Furthermore, the difference is significant for GPT-3.5 on relevance (p<0.001).
>
> ---
> > Beam size, # of model calls and performance:
>
> The search space of the candidate ranking is determined by both the beam size and the uncertainty threshold. A large beam size allows for more ranking permutations to be considered when the uncertainty threshold is fixed.
> We provide the following ablation study with an uncertainty threshold of 0.6 and reporting Spearman corr. along with the number of model calls in ():
>
> |Beam Size |Mistral-7B|Llama2-7B|
> |-|-|-|
> |1|22.9 (39.3)|13.5 (34.2)|
> |5|23.5 (73.7)|15.1 (70.5)|
> |20|23.8 (91.1)|16.6 (91.5)|
> |100|25.3 (103.8)|16.8 (94.6)|

---

> > ### Author Response · Authors · 2024-06-04
> > **Following up discussion towards the rebuttal**
> >
> > We thank the reviewer again for the insightful feedback. We also provide some further discussion on the performance on Larger LLMs:
> >
> > > Performance on Larger LLMs
> >
> > While computational resources prevent us from conducting larger-scale evaluations, we additionally include results from Llama2-13B, and we will endeavor to include larger-scale LLMs in future work in addition to the large-scale GPT4 experiments that we have conducted :
> >
> > ||PairS-greedy|
> > |-|-|
> > |Llama2-7B|15.3+-1.6|
> > |Llama2-13B|17.9+-1.1|

---

> > ### Author Response · Authors · 2024-06-05
> > **Reminder**
> >
> > Dear Reviewer VjHC,
> >
> > We thank you once again for your feedback! As the discussion period is ending, we wonder if you could take a look at our response if you haven't? We are eager to know whether our response has addressed your concerns: we'd be grateful if you could improve the rating if that's the case. Otherwise, we are always happy to answer further questions that you may have.
> >
> > Best,
> >
> > Authors

---

> > > ### Comment · Reviewer_VjHC · 2024-06-05
> > > **Thanks**
> > >
> > > Thank you to the authors for their response, the additional ablation on beam size and the result of different size are helpful. I am still curious of inference speed vs number of beam sizes but my questions have mostly been clarified.

---

> > > > ### Author Response · Authors · 2024-06-06
> > > > **Thanks**
> > > >
> > > > We thank the review for confirming all questions have been answered.
> > > >
> > > > As for the inference speed, the numbers in the bracket are averaged numbers of model forward calls. We believe it can represent the inference speed. The actual inference time depends on the LLM and GPU used. For example, in our case, it takes about 7s to perform a PairS-greedy (around 40 model calls) with Mistral-7B and batch size of 5, using RTX3090.

---

### Decision · Program_Chairs · 2024-07-10

**Decision:**

Accept

**Comment:**

This paper presents a new technique for using LLMs as evaluators, by formulating the evaluation problem as a ranking problem, and using LLMs to conduct pairwise comparisons to create a global ranking. LLM-based evaluation is a very popular technique being used in industry today and this work addressing how to improve it is very timely. Reviewers were positive about the clarity of writing, motivation, comprehensive experiments on open and closed source models (though larger models can be added), ablations and strong empirical results.

Reviewers mainly had minor clarification comments and some suggestions on presentation. The title of the paper may be slightly misleading because it mentions human judgement, while there is no direct comparison with human annotators in the paper, so the authors may want to revise it.

[comments from the PCs] As much as possible, please follow the AC request regarding the title to avoid confusing readers.